Development of a multiplex qPCR assay for the simultaneous detection of Mycoplasma bovis, Mycoplasma species, and Acholeplasma laidlawii in milk

Chauhan Kanika 1 2 3
http://orcid.org/0000-0003-0330-5013 Aly Sharif S. 1 2
http://orcid.org/0000-0003-0896-3939 Lehenbauer Terry W. 1 2
http://orcid.org/0000-0003-1456-9903 Tonooka Karen H. 1
Glenn Kathy 1
Rossitto Paul 1
Marco Maria L. 3 mmarco@ucdavis.edu
1 Veterinary Medicine Teaching & Research Center, School of Veterinary Medicine, University of California, Davis , Tulare, CA , United States
2 Department of Population Health and Reproduction, School of Veterinary Medicine, University of California, Davis , Davis, CA , United States
3 Department of Food Science and Technology, University of California, Davis , Davis, CA , United States
Gillespie Joseph
Electronic publication date: 2021 Aug 12
Publication date: 2021
Volume: 9
Electronic Location ID: e11881
Received 2021 Mar 22; Accepted 2021 Jul 7
Copyright: © 2021 Chauhan et al.
Copyright year: 2021
Copyright holder: Chauhan et al.
License: This is an open access article distributed under the terms of the Creative Commons Attribution License, which permits unrestricted use, distribution, reproduction and adaptation in any medium and for any purpose provided that it is properly attributed. For attribution, the original author(s), title, publication source (PeerJ) and either DOI or URL of the article must be cited.
License URL: https://creativecommons.org/licenses/by/4.0/

Keywords: Milk, Bovine mastitis, qPCR, Multiplex, Taqman, Diagnostic, Mycoplasma

Funding: The California Dairy Research Foundation (CDRF) P-15-003-UCD‐MM-SS USDA National Institute of Food and Agriculture Formula Funds CA-V-PHR-4042-H UC Davis School of Veterinary Medicine Dairy Herd Health & Food Safety Endowment CALV-DHHFS-0050 & CALV-DHHFS-0052 VMTRC, Tulare, CA and dairy owners The California Dairy Research Foundation (CDRF) supported this work (P-15-003-UCD‐MM-SS). Other support for this project was from the USDA National Institute of Food and Agriculture Formula Funds (Hatch project) (CA-V-PHR-4042-H) and the UC Davis School of Veterinary Medicine Dairy Herd Health & Food Safety Endowment (CALV-DHHFS-0050 & -0052). This work was also supported by the laboratory and technical staff at VMTRC, Tulare, CA and dairy owners who supplied milk samples that generated the isolates used in the study. The funders had no role in study design, data collection and analysis, decision to publish, or preparation of the manuscript.

==============================
Contagious bovine mastitis caused by Mycoplasma bovis and other Mycoplasma species including Mycoplasma californicum, Mycoplasma bovigenitalium, Mycoplasma alkalescens, Mycoplasma arginini, and Mycoplasma canadense is an economical obstacle affecting many dairy herds throughout California and elsewhere. Routine bacteriological culture-based assays for the pathogens are slow and subject to false-positive results due to the presence of the related, non-pathogenic species Acholeplasma laidlawii. To address the need for rapid and accurate detection methods, a new TaqMan multiplex, quantitative real-time PCR (qPCR) assay was developed that targets the 16S rRNA gene of Mycoplasma, rpoB gene of M. bovis, and the 16S to 23S rRNA intergenic transcribed spacer (ITS) region of A. laidlawii. qPCR amplification efficiency and range of detection were similar for individual assays in multiplex as when performed separately. The multiplex assay was able to distinguish between M. bovis and A. laidlawii as well as detect Mycoplasma spp. collectively, including Mycoplasma californicum, Mycoplasma bovigenitalium, Mycoplasma canadense, Mycoplasma arginini and Mycoplasma alkalescens. In milk, the lower limit of detection of M. bovis, M. californicum, and A. laidlawii with the multiplex assay was between 120 to 250 colony forming units (CFU) per mL. The assay was also able to simultaneously detect both M. bovis and A. laidlawii in milk when present in moderate (103 to 104 CFU/mL) to high (106 to 107 CFU/mL) quantities. Compared to laboratory culture-based methods, the multiplex qPCR diagnostic specificity (Sp) was 100% (95% CI [86.8–100]; n = 26) and diagnostic sensitivity (Se) was 92.3% (95% CI [74.9–99.1]; n = 26) for Mycoplasma species in milk samples collected from California dairy farms. Similarly, the Sp was 100% (95% CI [90.5–100]; n = 37) and Se was 93.3% (95% CI [68.1–99.8]; n = 15) for M. bovis. Our assay can detect and distinguish among M. bovis, other prevalent Mycoplasma spp., and non-pathogenic Acholeplasma laidlawii for effective identification and control of mycoplasma mastitis, ultimately supporting dairy cattle health and high-quality dairy products in California.

Introduction

Mastitis caused by Mycoplasma species is highly infectious in dairy cattle and causes serious disease and economic burdens, especially in large dairy herds in the U.S. and other countries worldwide (Nicholas, Fox & Lysnyansky, 2016). Pathogenic Mycoplasma spp. are unique compared to most other bacterial causes of mastitis because they lack a cell wall and have smaller genomes (Fox, Kirk & Britten, 2005). These organisms infect the udder tissue of dairy cows, and the resultant disease has long-term effects on milk quality, yield, and animal health (Nicholas, Fox & Lysnyansky, 2016). In the US alone, the mean cost per clinical mastitis case caused by Mycoplasma spp. and other microorganisms besides Gram-positive and Gram-negative bacteria was estimated at US $95.31 annually, with majority of the costs attributed to treatment (Cha et al., 2011). However, because of the very contagious nature and recognized poor response to therapy, dairy cattle with udder infections due to Mycoplasma spp. are often removed from the herd following identification of infection which significantly adds to the economic burden of mycoplasma mastitis (Ball, 1999; Maunsell et al., 2011; Nicholas, Fox & Lysnyansky, 2016). The most prevalent species causing cattle mastitis is Mycoplasma bovis, but other Mycoplasma species including Mycoplasma californicum and Mycoplasma bovigenitalium are also of diagnostic interest to dairy cattle farmers in California (Kirk et al., 1997; Infante-Martinez, Aguado & Eduard-Jasper, 1999; Brenner et al., 2009).

Infectious Mycoplasma spp. are highly contagious, frequently resistant to antibiotic treatment, and can result in chronic and subclinical states of infection. Because Mycoplasma can spread quickly, mastitis caused by Mycoplasma spp. is particularly detrimental to large (>500 cow) dairy herds (Jasper, 1982; Pfutzner & Sachse, 1996). For these reasons, screening of bulk tank milk or pooled animal milk samples is important to facilitate removal of infected animals thereby limiting the spread of infection. Routine Mycoplasma investigations involve using laboratory-culture based methods to enrich the bacteria from milk. This approach is challenging considering the numerous nutrient requirements and specific incubation conditions required for Mycoplasma growth. Even under optimal conditions, some mycoplasma strains are still difficult to grow on standard laboratory culture media (Razin, 1996). Other challenges include the extended incubation time (approximately 7 to 10 days) required before reaching sufficient cell numbers for analysis and also the time needed to identify Mycoplasma at the species level, usually by serological testing. Additionally, phenotypic similarities between Mycoplasma spp. and commensal environmental contaminants, most predominantly, Acholeplasma laidlawii, can result in false positives (Pfutzner & Sachse, 1996; Clothier et al., 2010; Fox, 2012).

Nucleic-acid based, culture-independent methods are now compelling alternatives for Mycoplasma diagnosis because of their potential to be rapid, accurate, and cost effective. Use of PCR and qPCR-based tests, in particular, have the greatest promise to be the most rapid and reliable methods for Mycoplasma spp. detection, identification, and quantification. To this regard, several end-point PCR and quantitative real-time PCR (qPCR) assays for M. bovis have been developed targeting M. bovis 16S rRNA (van Kuppeveld et al., 1992; Chavez Gonzalez et al., 1995; Cornelissen et al., 2017), uvrC (Clothier et al., 2010; Rossetti, Frey & Pilo, 2010; Naikare et al., 2015; Gioia et al., 2016; Behera et al., 2018), oppD (Sachse et al., 2010), fusA (Boonyayatra et al., 2012), and gltX (Appelt et al., 2019).

When assessing outbreaks of mycoplasma mastitis in dairies, especially for California’s large dairy herds, multiplex qPCR can be a favorable and practical approach for diagnostic testing. The use of different primer and probe sets in a single reaction tube significantly shortens the detection times and increases throughput capacity (Chapela, Garrido-Maestu & Cabado, 2015). A multiplex qPCR assay was developed to detect M. bovis, M. californicum, and M. bovigenitalium in milk samples with a limit of detection of between 102 to 105 colony forming units (CFU) per mL (Parker et al., 2017). There are also commercially-available, multiplex qPCR kits for milk such as PathoProof, Mastitis Major-3 kit (Thermo Fisher Scientific, Finland), and bactotype Mastits HP3 PCR Kit (Qiagen, Leipzig, Germany) designed to detect M. bovis along with the other contagious mastitis pathogens Staphylococcus aureus and Streptococcus agalactiae. However, compared to Mycoplasma spp., both S. aureus and S. agalactiae are relatively easy to isolate using conventional aerobic culture methods with final results available within 48 h.

In this study, we developed a multiplex qPCR assay with Taqman probes for detection of Mycoplasma spp., M. bovis, and A. laidlawii. The specificity and quantification range were verified for the individual primer sets as well as in multiplex. The multiplex assay was then tested for its diagnostic capability compared to routine laboratory, culture-based methods.

Materials and Methods

Bacterial strains and milk samples

M. bovis ATCC 25523, M. californicum ATCC 33461, M. bovigenitalium ATCC 19852, Mycoplasma canadense ATCC 29410, Mycoplasma arginine ATCC 23838, Mycoplasma alkalescens ATCC 29103 and A. laidlawii ATCC 23206 were purchased from the American Type Culture Collection (ATCC, Manassas, VA, USA) (Table 1). M. bovis ATCC 25523 and A. laidlawii ATCC 23206 genomic DNA (gDNA) was also purchased from the ATCC.

Table 1 Bacterial and algal strains used in this study.

Organism	Strain/Sourcea	
Mycoplasma bovis
Mycoplasma californicum	ATCC 25523 and field isolates (n = 3)
ATCC 33461 and field isolates (n = 3)	
Mycoplasma bovigenitalium	ATCC 19852 and field isolates (n = 3)	
Mycoplasma canadense	ATCC 29410 and field isolates (n = 3)	
Mycoplasma arginini	ATCC 23838 and field isolates (n = 3)	
Mycoplasma alkalescens	ATCC 29103 and field isolates (n = 3)	
Acholeplasma laidlawii	ATCC 23206 and field isolate (n = 3)	
Aerococcus viridans	ATCC 700406	
Bacillus cereus	MQ 17	
Enterobacter cloacae subsp cloacae	ATCC 13847	
Enterobacter sp.	MQ 14M0100-3	
Enterococccus faecalis	ATCC 29212	
Enterococcus durans	MQ 15M0470-1	
Enterococcus faecium	ATCC 35667	
Enterococcus gallinarium	ATCC 700425	
Escherichia coli	ATCC 25922	
Klebsiella pneumoniae	MQ 14	
Proteus hauseri	ATCC 13315	
Prototheca species	MQ 16M1258-4	
Pseudomonas aeruginosa	ATCC 10145	
Serratia marcescens	MQ 15M0394-3	
Staphylococcus aureus	MQ 16M1264-9; ATCC 25923	
Staphylococcus capitis subsp capitis	ATCC 35661	
Staphylococcus chromogenes	DFSL 8435	
Staphylococcus cohnii	ATCC 35662	
Staphylococcus epidermidis	DFSL 1780	
Staphylococcus haemolyticus	DFSL 8043	
Staphylococcus pasteuri	DFSL 8109	
Staphylococcus warneri	MQ 15M0945	
Staphylococcus xylosus	ATCC 35663	
Streptococcus agalactiae	ATCC 27956	
Streptococcus dysgalactiae subsp dysgalactiae	ATCC 27957	
Streptococcus dysgalactiae subsp equisimilis	ATCC 35666	
Streptococcus equi subsp zooepidemicus	ATCC 700400	
Streptococcus equinus	MQ 15M0757-12	
Streptococcus infantarius subsp coli	ATCC 27960	
Streptococcus mutans	MQ 15M1463-3	
Streptococcus uberis	MQ 16M1263-101	
Streptococcus uberis	ATCC 27958	
Streptococcus uberis	ATCC 700407	
Note:

a ATCC, American Type Culture Collection (Manassas, VA, USA); MQ, Milk Quality Laboratory (VMTRC; UC Davis, Tulare, CA, USA); DFSL, Dairy Food Safety Laboratory (VMTRC; UC Davis, Tulare, CA, USA).

Field isolates of Mycoplasma and Acholeplasma were obtained by routine bacteriological testing of milk samples submitted by producers and milk processors to the Milk Quality Laboratory (MQL) of the University of California, Davis Veterinary Medicine Teaching and Research Center (VMTRC, Tulare, CA, USA). The samples were from individual cows and bulk tanks located at dairies in the Central Valley, California (Table 1). All Mycoplasma and Acholeplasma isolates were recovered using VMTRC MQL protocols adapted from the National Mastitis Council (US, 2004). Briefly, milk samples were plated on modified mycoplasma agar (University of California Davis Veterinary Medicine Biological Media Services) directly upon receipt as well as after enrichment in mycoplasma specific broth (UC Davis VM Biological Media Services) for 48 h in 4% CO2 at 37 °C. Putative Mycoplasma and Acholeplasma colonies were streaked for isolation and subsequently identified to the species level by staining with fluorescent antibodies according to previously described methods (Baas & Jasper, 1972). To discriminate between A. laidlawii and Mycoplasma spp., a digitonin disk inhibition test was performed according to the method previously described (Thurmond, Holmberg & Luiz, 1989). All isolates were stored in 15% (v/v) glycerol at −80 °C.

Additionally, a total of 33 isolates from 30 different microbial species other than mycoplasmas and commonly found in bovine milk were used for cross-specificity evaluation of the qPCR assay (Table 1). These isolates were either obtained from the ATCC as type strains or isolated at the MQL and Dairy Food Safety Laboratory (DFSL) of University of California, Davis Veterinary Medicine Teaching and Research Center (VMTRC, Tulare, CA, USA) from routine diagnostic milk sample submissions from dairies located in the Central Valley, CA. Species identity was confirmed by 16S rRNA gene and rpoB sequencing (Drancourt et al., 2000; Drancourt & Raoult, 2002).

Genomic DNA extraction

The DNeasy Blood and Tissue Lysis kit (Qiagen, Carlsbad, CA, USA) was used for bacterial gDNA extraction and purification according to the manufacturer’s instructions. The gDNA was extracted from Mycoplasma and Acholeplasma strains after incubation in mycoplasma specific broth (UC Davis VM Biological Media Services) at 37 °C in 4% CO2 for between 24 to 72 h (Appelt et al., 2019). The gDNA from other microbial species (Table 1) was isolated from colonies retrieved from bovine blood agar plates (UC Davis VM Biological Media Services) after incubation at 37 °C for 24 to 48 h. The commercial kit was used for the isolation of gDNA in order to support the elimination of PCR inhibitors that affect the sensitivity of the assay or even lead to false negative results (Schrader et al., 2012). For gDNA extractions from bacteria contained in milk, 1 ml milk was centrifuged at 18,407×g for 10 min. The resulting pellet was then suspended in Buffer ATL (Qiagen, Carlsbad, CA, USA) and Proteinase K (Qiagen, Carlsbad, CA, USA) for incubation at 56 °C for 1 h with continuous mixing prior to DNA extraction and purification. The gDNA concentrations were measured on a Qubit 3.0 Fluorometer using the Qubit dsDNA HS Assay Kit (Life Technologies, Eugene, OR, USA). All gDNA was stored at −20 °C until analysis or at 4 °C for no longer than 2 to 4 weeks.

qPCR primer design

Clustal W (Thompson, Higgins & Gibson, 1994) was used to identify conserved and variable Mycoplasma and Acholeplasma DNA targets for the development of new primer-probe pairs to be used for the qPCR assays. DNA sequence alignments were performed for 16S rRNA, rpoB and intergenic transcribed spacer (ITS) region between 16S and 23S rRNA genes in M. bovis, M. californicum, M. bovigenitalium, M. canadense M. alkalescens, and A. laidlawii. Those Mycoplasma species were selected for primer development because of their prevalence in Mycoplasma mastitis reported in the US dairy herds (Fox, 2012; Nicholas, Fox & Lysnyansky, 2016) and by the VMTRC MQL. The following strains were used for primer selection: M. bovis strain PG45 (ATCC 25523, NC_014760) (Wise et al., 2011), M. californium strain ST-6 (ATCC 33461, NZ_CP007521) (Calcutt, Foecking & Fox, 2014), M. bovigenitalium strain HAZ (ATCC 19852, AP017902) (Edward & Freundt, 1956), M. canadense strain HAZ 360_1 (ATCC 29410, NZ_AP014631), M. alkalescens 14918 (ATCC 29103, NZ_AMWK01000000), and A. laidlawii PG-8A (ATCC 23206, NC_010163) (Lazarev et al., 2011). DNA sequence alignments showed sufficient nucleotide variation in the 16S rRNA and rpoB genes and ITS region for primer and Taqman probe design (Fig. S1).

All qPCR primers and Taqman probes were designed using Primer-BLAST and Primer Express software (Applied Biosystems, Waltham, MA, USA) (Table 2 and Fig. S1). Primers were regarded to be acceptable if they contained at least two total mismatches to unintended targets, including at least two mismatches within the last five base pairs of the 3′ end. All primers were designed to have a melting temperature at 59 °C to 62 °C and an amplicon size between 108 to 232 bp.

Table 2 Sequences of primers, probes, PCR product sizes and amplification efficiency of the qPCR assaysa.

Organism	Target	Primer and probeb	Primer and probe sequence (5′ to 3′)	nM	Product (bp)	qPCR efficiency (%)c	
Singleplex	Multiplex	
Mycoplasma sppd	16S rRNA	Myco_F
Myco_R
Myco_Probe	CGAGCGCAACCCTTATCCTT
CCCCACTCGTAAGAGGCATGA
VIC-TCGTCCCCACCTTCCTCCCG-QSY	100
100
75	118	88.8–100.9	96.8–102.4	
M. bovis	rpoB	M.bovis_F
M.bovis_R
M.bovis_Probe	TTTCAGCCGCTAACTTCAGAGC
GCAAGTTCCCCATCCTTGAAG
ABY-TCGCCTTTAGCAACTTCTTGACCAA-QSY	200
200
200	232	87.1	95.6	
A. laidlawii	ITS	Achol_F
Achol_R
Achol_Probe	AAGTGGGCAATACCCAACGC
ACGTTCCCGTAGGGATACCTTG
6-FAM-ACGGCTCCCTCCCTTTCGGG-QSY	15015075	108	91.9	91.2	
Notes:

a The QuantiFast Multiplex PCR master mix (Qiagen, Redwood City, CA, USA) was used for all assays and with an annealing temperature of 58 °C.

b F and R indicate forward and reverse primers, respectively. TaqMan probes were designed with 6-FAM (6-carboxyfluorescein) VIC (2’-chloro-7phenyl-1,4-dichloro-6-carboxy-fluorescein) and ABY (Thermo Fisher, Waltham, MA, USA) as reporter dyes on the 5′ end and QSY 7 (QSY7 succinimidyl ester) as the quencher dye on the 3′ end.

c Amplification efficiencies were determined using gDNA from M. bovis ATCC25523, M. californicum ATCC 33461, and M. bovigenitalium ATCC19852 (16S rRNA), M. bovis ATCC25523 (rpoB), and A. laidlawii ATCC 23206 (ITS).

d The PCR amplification efficiencies for M. bovis, M. californicum ATCC 33461, and M. bovigenitalium ATCC19852 were measured as 88.8%, 97.1%, and 100.9% in singleplex and 96.8, 102.4% and 97.9% in multiplex, respectively.

qPCR assay parameters

All primers and Taqman probes were optimized by testing concentrations in the range of 100 nM to 400 nM and 75 nM to 400 nM, respectively. qPCR amplification was performed in 96-well plates on an Applied Biosystems 7500 Fast thermal cycler (Applied Biosystems, Carlsbad, CA, USA) with an initial denaturation step of 5 min at 95 °C followed by 40 cycles of 45 s at 95 °C for DNA denaturation and 45 s at 58 °C for primer and probe hybridization and DNA elongation steps. The multiplex assay, combining the three individual assays for Mycoplasma, M. bovis, and A. laidlawii, was carried out in a total volume of 20 µl, comprising 5 µl of the gDNA, 10 µl of 2X QuantiFast Multiplex PCR master mix (Qiagen, Redwood City, CA, USA), and between 75 and 200 nM of each primer and TaqMan probe (Table 2). qPCR assay efficiencies were calculated using standard curves based on 10-fold serial dilutions of gDNA from M. bovis ATCC 25523 for the rpoB and 16S rRNA assays and A. laidlawii ATCC 23206 for the ITS assay.

Quantitative ranges of the qPCR assays

Serial dilutions were prepared of gDNA from M. bovis ATCC 25523, M. californicum ATCC 33461, M. bovigenitalium ATCC 19852, and A. laidlawii ATCC 23206. The serial dilutions spanned a 106-fold range, encompassing between approximately 5 fg to 5 ng gDNA. The QuantiFast Multiplex PCR Master mix was used in combination with the primer and Taqman probe concentrations described in Table 2 with an annealing temperature of 58 °C. The assays were tested in singleplex and multiplex in triplicate and in two separate runs and for each gDNA primer/probe combination. To estimate genome equivalents, it was assumed that the average base pair weighs 650 g/mol and the average genome sizes were 1 Mbp for M. bovis (Wise et al., 2011), 0.79 Mbp for M. californicum (Calcutt, Foecking & Fox, 2014), 0.86 Mbp for M. bovigenitalium (Hata, Nagai & Murakami, 2017), and 1.5 Mbp for A. laidlawii (Lazarev et al., 2011). For verifying assay specificity, a total of 1 to 25 ng gDNA from diverse microbial strains (Table 1) was tested for detection.

To examine the quantitative range of the multiplex assay in milk, raw bovine milk samples were first confirmed to be free of Mycoplasma spp. by plating on modified mycoplasma agar and incubation at 37 °C in 4% CO2 for 7 days. Next, 1 × 106 to 1 × 108 colony-forming units (CFU)/mL of M. bovis ATCC 25523, M. californicum ATCC 33461, and A. laidlawii ATCC 23206 were inoculated into separate aliquots of that milk. The cell suspensions were immediately used for serial dilutions in the same raw milk so that the final cell numbers spanned a 104-fold (M. californicum) to 106-fold (M. bovis and A. laidlawii) range. gDNA was then extracted and tested with the multiplex assay using conditions described in Table 2 before setting multiplex qPCR assay detection criteria.

Table 3 Criteria for detection of Mycoplasma spp., M. bovis, and A. laidlawii in milk with the multiplex qPCR assay.

Organismb	Gene target (Cut-off Ct value)a	
16S rRNA
(Ct ≤ 32)	rpoB
(Ct ≤ 33)	ITS
(Ct ≤ 32)	
Mycoplasma spp.	+	–	–	
M. bovis c	+	+	–	
A. laidlawii	–	–	+	
Notes:

a Ct, Threshold Cycle value; positive (+) and negative (−) symbols indicate a positive and negative result for that group of bacteria by the multiplex qPCR assay.

b M. californicum ATCC33461 M. bovis ATCC25523, and A. laidlawii ATCC23206 were used for determining LOD levels of Mycoplasma, M. bovis, and A. laidlawii, respectively.

c M. bovis gDNA was expected to be detected by both the 16S rRNA and rpoB assays, therefore a positive M. bovis sample is expected to have <3 Ct value difference for those targets.

Multi-species detection

Mixtures of M. bovis ATCC 25523 and A. laidlawii ATCC 23206 gDNA were tested with the multiplex assay to assess for the capacity for simultaneous, multi-species detection. gDNA was mixed instead of intact cells because growth rate variations between the strains limited our capacity to obtain sufficient cell numbers within the same period of time. To prepare the gDNA mixtures, gDNA was extracted from a single strain immediately after inoculation into separate aliquots of raw milk at either low (102 to 103 CFU/mL), medium (103 to 104 CFU/mL), or high (106 to 107 CFU/mL) cell numbers. Next, gDNAs from those strains were mixed in equal volumes (1:1 ratios) and tested with the multiplex assay using the assay conditions described in Table 2. A total of nine combinations of gDNAs, representing low, medium, and high levels of M. bovis and A. laidlawii cells were measured.

Table 4 Detection of M. bovis and A. laidlawii mixtures in milk with the multiplex qPCR assaya.

		Target	Ct (avg ± stdev)b	
M. bovis (low)	A. laidlawii (low)	16S rRNA	33.47 ± 0.24	
A. laidlawii (intermediate)		33.32 ± 0.12	
A. laidlawii (high)		33.41 ± 0.41	
A. laidlawii (low)	rpoB	33.80 ± 0.52	
A. laidlawii (intermediate)		33.47 ± 0.21	
A. laidlawii (high)		33.49 ± 0.66	
A. laidlawii (low)	ITS	32.91 ± 0.38	
A. laidlawii (intermediate)		25.65 ± 0.07	
A. laidlawii (high)		15.79 ± 0.11	
M. bovis (intermediate)	A. laidlawii (low)	16S rRNA	27.44 ± 0.11	
A. laidlawii (intermediate)		27.42 ± 0.12	
A. laidlawii (high)		27.22 ± 0.16	
A. laidlawii (low)	rpoB	27.05 ± 0.23	
A. laidlawii (intermediate)		26.97 ± 0.10	
A. laidlawii (high)		26.83 ± 0.31	
A. laidlawii (low)	ITS	32.87 ± 0.51	
A. laidlawii (intermediate)		25.71 ± 0.07	
A. laidlawii (high)		15.77 ± 0.07	
M. bovis (high)	A. laidlawii (low)	16S rRNA	16.68 ± 0.15	
A. laidlawii (intermediate)		16.63 ± 0.18	
A. laidlawii (high)		16.90 ± 0.05	
A. laidlawii (low)	rpoB	16.38 ± 0.19	
A. laidlawii (intermediate)		16.42 ± 0.22	
A. laidlawii (high)		16.58 ± 0.14	
A. laidlawii (low)	ITS	36.70 ± 0.71	
A. laidlawii (intermediate)		26.02 ± 0.15	
A. laidlawii (high)		15.69 ± 0.05	
Notes:

a gDNA was extracted from milk inoculated with M. bovis ATCC 25523 and A. laidlawii ATCC 23206 at either low (102 to 103 CFU/mL), medium (103 to 104 CFU/mL), or high (106 to 107 CFU/mL).

b Shading indicates average Ct values within range for detection (n = 3) using qPCR criteria for multiplex assay given in Table 3.

qPCR performance on field milk samples

To compare the qPCR assay performance to conventional culture-based methods, a total of 52 frozen (−20 °C) bulk tank and individual cow milk samples were first confirmed to be either Mycoplasma spp. culture positive or negative. This was determined by plating the milk samples directly on modified mycoplasma agar as well as after enrichment in mycoplasma specific broth for 24 h in 4% CO2 at 37 °C. Colonies were evaluated by the same person at 4 and 7 days and verified as M. bovis, M. bovigenitalium, or M. alkalescens using staining with fluorescent antibodies according to previously described methods (Baas & Jasper, 1972). The 52 milk samples were then anonymized for testing by the multiplex qPCR assay and 5 µl of total DNA extracted from the milk was tested using assay parameters described in Table 2. Diagnostic sensitivity (Se) and diagnostic specificity (Sp) of the multiplex qPCR assay was calculated as described previously (Gioia et al., 2016). Using the results from laboratory culture as the reference, the number of true positive (TP), true negative (TN), false positive (FP), and false negative (FP) samples were combined into following formulas: Se (%) = TP/(TP + FN) and Sp (%) = TN/(TN + FP). Kappa, an estimate of agreement beyond chance between laboratory culture and qPCR was estimated and interpreted as <40%, poor; 41 to 75%, fair to good; >75%, excellent (Fleiss, Levin & Paik, 2003).

Results

qPCR assay efficiency and range of detection for target gDNA

The Mycoplasma 16S ribosomal RNA (16S rRNA) gene, M. bovis rpoB, and the A. laidlawii ITS region were selected for qPCR primer design. These targets were previously shown to be useful markers for detection of Mycoplasma and Acholeplasma in milk and clinical samples (van Kuppeveld et al., 1992; Cornelissen et al., 2017; Gioia et al., 2016; Parker et al., 2017; Tang et al., 2000). After testing a range of concentrations for each primer and TaqMan probe, the Taqman qPCR assay parameters (QuantiFast Multiplex PCR master mix; annealing temperature of 58 °C) and assay primer and probe concentrations were set as described in Table 2. The qPCR amplification efficiencies for each of the individual qPCR assays on intended target gDNA ranged from 90% to 102.4% when tested separately (in singleplex) (Table 2 and Fig. S2) and combined into one Mycoplasma-M. bovis-A. laidlawii multiplex assay (Table 2 and Fig. 1). For the 16S rRNA gene assay in multiplex, the qPCR amplification efficiency was high for M. bovis (Table 2) (96.8%) and comparable to efficiencies found for M. californicum ATCC 33461 (102.4%) and M. bovigenitalium ATCC19852 (97.9%). The correlation coefficient of the straight line, R2 was higher than 0.98 for all assays over a 106-fold range in singleplex (Fig. S2) and multiplex (Fig. 1). The lower limit of detection (LLD) for each assay was similar between the singleplex (Fig. S2) and multiplex (Fig. 1) formats (Ct values from 30 to 35). Both formats were able to detect low numbers of DNA copies per PCR (50 fg DNA, approximately equal to 30 to 58 genome equivalents for the respective species being targeted by the assay).

Figure 1 Standard curves for the 16S rRNA, rpoB and ITS TaqMan qPCR assays performed in multiplex.

The standard curves were constructed with 10-fold serial dilutions of M. bovis ATCC 25523 (16S rRNA and rpoB assays), M. bovigenitalium ATCC 19852 (16S rRNA assay), M. californicum ATCC 33461 (16S rRNA assay), and A. laidlawii ATCC 23206 (ITS assay) gDNA, ranging from between approximately 5 fg to 5 ng gDNA. Results shown are from a single run with each dilution tested in triplicate. The R2 value was 0.99 for the standard curve of each target (16S rRNA, rpoB and ITS). Error bars indicate standard deviation (±) based on the results for three replicates.

Specificity of the multiplex qPCR assay

We next examined the qPCR assays for cross-reactivity between the other species targets contained in the Mycoplasma-M. bovis-A. laidlawii multiplex test. No amplification (Ct > 37) was found for the A. laidlawii 16-23S ITS assay when tested on either M. bovis ATCC25523, M. californicum ATCC33461, or M. bovigenitalium ATCC19852 gDNA template (data not shown). For the M. bovis rpoB assay, some amplification (Ct values 31 to 37) was observed for M. californicum ATCC 33461 and M. bovigenitalium ATCC 19852 gDNA. However, that Ct range of detection was approximately 17 Ct values higher than observed for equivalent quantities of the M. bovis target (0.5 ng), comparable to a 105 to 106-fold difference in detection sensitivity. For the Mycoplasma 16S rRNA assay, some amplification (Ct values 31 to 35) was observed for A. laidlawii ATCC 23206 gDNA at the highest quantity of gDNA tested (0.5 ng). However, just as found for the rpoB assay, those Ct values were also much higher (Ct difference of 17) than when A. laidlawii was used as the target, comparable to a 105 to 106-fold difference in detection.

No amplification (Ct > 37) was observed with the multiplex assay for any non-Mycoplasma strains from 33 different microbial species, including representatives of the Staphylococcus, Bacillus, Aerococcus, Enterococcus, Enterobacter, Escherichia, Klebsiella, Proteus, Prototheca, Pseudomonas, Serratia, and Streptococcus genera (Table 1). As expected, gDNA from M. californicum, M. bovigenitalium, M. canadense, M. arginini, and M. alkalescens isolates (Table 1) resulted in amplification with the 16S rRNA assay (Ct values < 28) but not with the rpoB and ITS assays (Ct values > 37). Moreover, gDNA from A. laidlawii field isolates only resulted in amplification with the ITS assay, while three other field isolates of M. bovis from milk were accurately detected by the rpoB and 16S rRNA assays (Ct values ranged from between 21 to 24).

Limit of Detection (LOD) of multiplex qPCR assay in milk

Next, the multiplex qPCR assay range of detection was tested for Mycoplasma and A. laidlawii in milk. A DNA amplification efficiency of 95% to 106% (R2 > 0.98) was found for all three gene targets spanning a 104 to 106-fold range when the multiplex assay was applied on gDNA extracted from serial dilutions of M. californicum ATCC33461, M. bovis ATCC25523, or A. laidlawii ATCC23206 contained in milk (Fig. 2). However, it was noted that milk alone (no-template control or mycoplasma culture negative) sometimes resulted in some DNA amplification (Ct cycles 33 to 37). Therefore, Ct values above 33 were reported as negative for the multiplex qPCR assay. To avoid false positives, a LOD cut-off Ct value was set at <32 for Mycoplasma, <33 for M. bovis, and <32 for A. laidlawii (Table 3). Those cut-off Ct values for each target assay were selected based on a minimum 3-Ct difference between the lowest quantities of the target organism in milk and the negative control Ct. Based on cell numbers resulting in that Ct, the LOD was within the range of 120 to 250 CFU/mL.

Figure 2 Limit of detection of the multiplex qPCR assay for M. bovis, M. californicum, and A. laidlawii in milk.

Serial dilutions of M. bovis ATCC 25523 (starting cell number of 2.5 × 108 CFU/mL), M. californicum ATCC 33461 (starting cell number of 2.3 × 106 CFU/mL), and A. laidlawii ATCC 23206 (1.22 × 108 CFU/mL) were performed in raw milk confirmed to be negative for Mycoplasma and A. laidlawii prior to gDNA extraction and target detection with the multiplex qPCR assay. Results shown are from a single run with each dilution tested in duplicate. The R2 value was 0.99 for the standard curve of each target (16S rRNA, rpoB and ITS). Error bars indicate standard deviation (±) based on the results for all replicates.

Multi-species detection with the multiplex qPCR assay

Application of the multiplex qPCR assay on gDNA from M. bovis ATCC 25523 and A. laidlawii ATCC 23206 mixed in different proportions showed that multiplex assay could simultaneously detect M. bovis and A. laidlawii present in the same sample (Table 4). The 16S rRNA and rpoB targeted assays were positive when gDNA from milk containing either moderate (103 to 104 CFU/mL) to high (106 to 107 CFU/mL) numbers of M. bovis was applied, irrespective of A. laidlawii gDNA abundance. Similarly, the A. laidlawii ITS qPCR assay results were positive when gDNA from milk containing moderate or high numbers of A. laidlawii was used, even in the presence of high quantities of M. bovis gDNA. Neither organism was detectable according to qPCR when the gDNA was extracted from milk with low cell numbers (102 to 103 CFU/mL) (Table 4).

qPCR performance on field milk samples

To determine the extent to which the multiplex qPCR assay can reliably detect and differentiate between M. bovis from non-bovis Mycoplasma spp. and A laidlawii, a total 52 milk samples collected from bulk tanks and individual cows on CA dairy farms were analyzed with the assay and the results were compared to laboratory culture-based detection methods (Table 5). No sample was shown to contain A. laidlawii and therefore the presence of that organism in field milk samples was not evaluated. Using the Mycoplasma spp. qPCR Ct value criteria we set for the multiplex assay (Table 3), no false-positive results, Sp 100% (95% CI [86.8–100]; n = 26) were obtained for the milk samples which were also found to lack Mycoplasma spp. according to laboratory culture-based methods (Table 5). Similarly, using the M. bovis criteria (Table 3), no false positive results, Sp 100% (95% CI [90.5–100]; n = 37) were obtained for the milk samples found to lack M. bovis according to laboratory culture-basedmethods (Table 5). Notably, the assay parameters were discriminative because even though a sample containing M. alkalescens (ID# 482582) yielded a Ct value of approximately 32 for the M. bovis qPCR assay, there was an approximate five-Ct difference between the rpoB and 16S rRNA assays. That difference was beyond the allowable limit for M. bovis by the multiplex assay, and therefore was appropriately identified as having Mycoplasma spp. and not M. bovis. (Table 3).

Table 5 Laboratory culture and multiplex qPCR results for field milk samples.

		Laboratory Cultureb	Multiplex qPCR Result (Ct)c	
ID#	Sample Typea	DP	MB	Organism	Gene Target	Organism	
16S rRNA	rpoB	ITS	
4750	C	≥400	+	M. bovis	+(15.10)	+(17.18)	–	M. bovis	
1331-2	T	4	+	M. bovis	+(29.79)	+(31.43)	–	M. bovis	
1985-1	T	85	+	M. bovis	+(24.60)	+(25.90)	–	M. bovis	
2927-3	T	2	–	M. bovis	+(31.03)	−(33.76)	–	Mycoplasma sp.	
3145-1	T	+	+	M. bovis	+(27.70)	+(32.35)	–	M. bovis	
3606-1	T	+	+	M. bovis	+(26.97)	+(28.34)	–	M. bovis	
3642-1	T	≥100	+	M. bovis	+(25.16)	+(27.80)	–	M. bovis	
3915-1	T	+	+	M. bovis	+(26.17)	+(27.32)	–	M. bovis	
43722-1	T	≥200	+	M. bovis	+(23.19)	+(24.91)	–	M. bovis	
43897-1	T	4	+	M. bovis	+(28.81)	+(31.46)	−(34.4)	M. bovis	
4395-1	T	≥600	+	M. bovis	+(26.27)	+(27.55)	–	M. bovis	
43967-2	T	1	+	M. bovis	+(29.42)	+(31.45)	−(34.52)	M. bovis	
4435-1	T	9	+	M. bovis	+(29.44)	+(30.62)	−(36.74)	M. bovis	
4437-1	T	≥200	+	M. bovis	+(23.94)	+(24.95)	−(35.99)	M. bovis	
4784-1	T	9	+	M. bovis	+(27.63)	+(29.77)	−(34.96)	M. bovis	
8229	C	–	+	M. bovigenitalium	+(17.84)	−(34.89)	−(37.21)	Mycoplasma sp.	
180	C	–	+	M. alkalescens	+(31.16)	–	–	Mycoplasma sp.	
6141	C	–	+	M. alkalescens	+(29.31)	−(38.47)	–	Mycoplasma sp.	
33257	C	–	+	M. alkalescens	+(26.78)	−(35.35)	–	Mycoplasma sp.	
39494	C	–	+	M. alkalescens	+(25.85)	−(38.41)	–	Mycoplasma sp.	
41672	C	–	+	M. alkalescens	–	–	–	–	
42275	C	6	+	M. alkalescens	+(25.25)	−(36.52)	−(34.61)	Mycoplasma sp.	
43960	C	–	+	M. alkalescens	−(34.67)	–	−(32.43)	–	
482582	C	1	+	M. alkalescens	+(26.46)	−(31.54)	–	Mycoplasma sp.	
749	C	85	+	M. alkalescens	+(19.99)	−(35.69)	–	Mycoplasma sp.	
750	C	–	+	M. alkalescens	+(26.76)	−(34.34)	–	Mycoplasma sp.	
829	C	–	–	–	−(32.42)	−(34.28)	−(34.89)	–	
818	C	–	–	–	−(33.07)	−(35.25)	−(35.56)	–	
802	C	–	–	–	−(33.18)	−(33.88)	–	–	
828	C	–	–	–	−(33.43)	−(34.23)	−(34.07)	–	
809	C	–	–	–	−(33.57)	−(34.22)	–	–	
819	C	–	–	–	−(34.19)	−(34.09)	−(34.71)	–	
815	C	–	–	–	−(34.54)	–	–	–	
810	C	–	–	–	−(34.57)	−(34.58)	–	–	
826	C	–	–	–	−(34.83)	−(34.35)	−(36.11)	–	
823	C	–	–	–	−(34.96)	–	−(33.46)	–	
804	C	–	–	–	−(35.26)	–	−(36.09)	–	
822	C	–	–	–	−(35.40)	–	–	–	
814	C	–	–	–	−(35.85)	–	–	–	
824	C	–	–	–	−(36.02)	−(38.96)	−(35.90)	–	
801	C	–	–	–	−(36.03)	–	–	–	
808	C	–	–	–	−(36.06)	−(34.92)	−(35.19)	–	
803	C	–	–	–	−(36.17)	−(39.70)	–	–	
820	C	–	–	–	−(36.94)	–	−(34.57)	–	
812	C	–	–	–	−(37.08)	−(34.66)	–	–	
817	C	–	–	–	−(37.37)	–	–	–	
806	C	–	–	–	−(37.65)	−(35.19)	−(34.94)	–	
811	C	–	–	–	−(37.90)	–	–	–	
813	C	–	–	–	−(38.00)	−(35.11)	−(34.94)	–	
821	C	–	–	–	−(38.25)	–	–	–	
807	C	–	–	–	–	–	−(34.88)	–	
816	C	–	–	–	–	–	−(34.84)	–	
Notes:

a Cow (C) or Bulk Tank (T) milk sample.

b DP, Direct plating. Based on results from direct plating of milk samples on modified mycoplasma agar using cotton swabs; MB, Mycoplasma Broth. Based on culture results of 24 h enriched broth samples; Plates with bacterial growth are indicated by a plus (+) and no growth by a minus (−); Mycoplasma spp. were identified by fluorescently labeled, species-specific antibody staining method. The organism column is shaded gray to show the species identified by laboratory culture based methods.

c The positive qPCR results are indicated by a plus (+) and negative results by a minus (−); Mycoplasma spp. were identified by Ct cut-off criteria (Table 3). The organism column is shaded gray to show the species was identified by the multiplex qPCR assay.

A total of 24 out of 26 milk samples that tested positive for Mycoplasma spp. according to CFU isolation in laboratory culture medium were also positive according to qPCR, Se 92.3% (95% CI [74.9–99.1]; n = 26). The remaining two samples (ID# 41672 and ID# 43960) were negative by qPCR for all three gene targets but tested positive for M. alkalescens after enrichment and plating for CFU identification (Table 5).

The multiplex assay also accurately detected 14 out of 15 milk samples determined to contain M. bovis according to culture-based methods, yielding a Se of 93.3% (95% CI [68.1–99.8]; n = 15) for M. bovis. The remaining milk sample (ID# 2927-3) tested positive for Mycoplasma spp. but not M. bovis (Table 5). That sample resulted in only two presumptive M. bovis colonies (approximately 100 CFU/ml) (Table 5). Notably the average Ct for that sample (Ct of 33.76) was only slightly above the upper Ct value cut-off for M. bovis (Ct ≤ 33) (Table 3) and the Ct for Mycoplasma detection with the 16S rRNA gene was near to the limit of the acceptable range (Ct of 31.03 compared to Ct < 32).

Both assays, laboratory culture and qPCR, had excellent agreement beyond chance as measured by Kappa. Specifically, the agreement between laboratory culture and qPCR in detecting milk samples testing positive for Mycoplasma spp. was 92.31% (SE 13.83; P value < 0.01) and for M. bovis was 95.22% (SE 13.85; P value < 0.01).

Discussion

The TaqMan multiplex qPCR assay developed here was shown to be a rapid and reliable diagnostic tool for detection of Mycoplasma spp. and M. bovis and discrimination between Mycoplasma and A. laidlawii in milk. The assay was accurate and showed a large dynamic range for quantification (106 fold range) while maintaining a comparable sensitivity to that of the individual qPCR assays for each target. Simultaneous monitoring was achieved for M. bovis and A. laidlawii and the diagnostic sensitivity and field performance of the multiplex qPCR assay was confirmed. The assay is a significant advancement for mastitis diagnostics because it emphasizes M. bovis detection while also affording the possibility to monitor for other Mycoplasma genera and simultaneous differentiation of non-pathogenic A. laidlawii contaminants.

The individual qPCR assays used here target genes that have also previously shown to be useful for Mycoplasma and A. laidlawii sensing by other methods. Conventional PCR assays targeting Mycoplasma 16S rRNA genes were used for detection (Hirose et al., 2001) and both detection and differentiation of different Mycoplasma spp., including M. bovigenitalium and M. californicum (McAuliffe et al., 2005). For developing genus- specific primers, target sequences used should have the greatest sequence divergence between genera but be conserved for species within the same genus. The 16S rRNA gene is an ideal target in this regard since the presence of two gene copies also increases the detection limit by PCR. In comparisons between the 16S rRNA genes of Mycoplasma and other related species including M. californicum, M. bovigenitalium, M. alkalescens, M. arginini and M. canadense, we observed significant sequence similarity in the V7 variable region. This finding agreed with a previous study which concluded that mycoplasma genus-specific sequences occur between the 16S rRNA V6 and V7 regions, although some cross reactivity was found with Acholeplasma species (van Kuppeveld et al., 1992). The alignment of the V7 regions showed a 118-bp sequence which distinguished A. laidlawii from Mycoplasma spp., while was also conserved among Mycoplasma species.

A variety of M. bovis genes including, uvrC, gltX, fusA and oppD have been targets for species detection by PCR methods (Clothier et al., 2010; Rossetti, Frey & Pilo, 2010; Sachse et al., 2010; Boonyayatra et al., 2012; Naikare et al., 2015; Gioia et al., 2016; Behera et al., 2018; Appelt et al., 2019). We selected rpoB, the gene encoding the highly conserved beta-subunit of the bacterial RNA polymerase, for assay development because there were more nucleotide variations in this gene between M. bovis and other Mycoplasma species. rpoB was also found to be a suitable target for phylogenetic analysis of Mycoplasma spp. including M. bovis (Kim et al., 2003). The use of the rpoB gene for M. californicum detection in milk has already been explored in a probe-based, qPCR assay (Boonyayatra et al., 2012; Parker et al., 2017). Therefore, in this study, we used a 232-bp sequence in rpoB gene for the design of primers and a Taqman probe that could detect M. bovis. Notably, this assay reached a higher amplification efficiency than the qPCR assay we previously developed for M. bovis targeting gltX (Appelt et al., 2019).

For the detection of A. laidlawii, the ITS region (108-bp; separating the two 16S–23S rRNA operons in A. laidlawii) was used for species-specific primers and Taqman probe development. The ITS region was previously shown to have sufficient heterogeneity to differentiate between Mycoplasma spp. and A. laidlawii (Tang et al., 2000; Gioia et al., 2016). This finding is consistent with the lack of amplification observed for M. bovis or the other Mycoplasma tested with that assay.

The individual qPCR assays performed well when combined in multiplex qPCR, with no observed cross-reactivity between primer pairs and impacts on the dynamic range of target detection. The multiplex assay was also capable of measuring gDNA from as few as 210 to 250 CFU/mL in milk for each of the targets. The sensitivity of this qPCR method is comparable to other assays which reported a detection limit for M. bovis in a range of between 10 and 240 CFU/mL in milk (Clothier et al., 2010; Sachse et al., 2010; Boonyayatra et al., 2012). Our assay was also more sensitive than the LOD reported for a multiplex qPCR assay targeting uvrC in M. bovis, rpoB in M. californicum, and the ITS region in M. bovigenitalium (Parker et al., 2017). In that study, the detection limit was 130 CFU/mL, 600 CFU/mL, and 105 CFU/mL for M. bovis, M californicum, and M. bovigenitalium, respectively. Sensitivity decreased by 100- to 1,000-fold when all three species were present at the same time to simulate a multispecies infection (Parker et al., 2017). Our prior qPCR assay and methods targeting the M. bovis gltX was less sensitive to detect low quantities of M. bovis cells. The detection limit was between 10 to 100 gDNA copies per reaction, corresponding to 104 to 105 cells per mL (Appelt et al., 2019). However, it should be noted that a rapid DNA extraction method was used along with other Taq polymerase reagents which may have impacted estimates of M. bovis detection limits.

Recently, a survey study performed on 95 New York dairy farms detected both M. bovis and A. laidlawii in the same sample in 6% of the farms and mixed mycoplasma infections in 14% of the farms (Gioia et al., 2016). Multispecies detection could be useful when testing for mastitis because the occurrence of mixed mycoplasma spp. infection has been reported previously, with M. bovis being isolated in approximately 90% of those cases and other species of mycoplasma isolated as secondary pathogens or contaminants (Kirk et al., 1997; Szacawa et al., 2015). Our multiplex qPCR assay reliably detected M. bovis and A. laidlawii gDNA when mixed together. This indicated that M. bovis would be detectable, even with high levels of A. laidlawii contaminants. Future studies will investigate the sensitivity of the multiplex qPCR assay for mixed Mycoplasma species present in milk samples.

The multiplex assay showed a high level of Sp and Se when tested on milk samples collected from different dairy farms in CA. Multiplex qPCR yielded a Se of 92.3% for 16S rRNA and 93.3% for rpoB and a Sp of 100% for each of the targets compared with milk samples confirmed by traditional laboratory bacterial culture methods. These results are comparable to those obtained by others examining for M. bovis in milk (Clothier et al., 2010; Parker et al., 2017; Appelt et al., 2019). Even though the sample size for the field validation test was small, no false positive results were obtained. Instead, such samples were correctly identified as negatives by qPCR as they did not meet the assay criteria for species/genera identification even when culture results were positive. We suggest that qPCR negative results can be associated with either the presence of PCR inhibitors or low Mycoplasma cell numbers which might have led to a general decrease in PCR sensitivity. In the former case the utilization of a DNA internal control in the same tube as the target can eliminate false-negative PCR results caused by PCR inhibiting substances. In the latter case, the multiplex assay cannot be used for direct detection in milk samples containing bacterial numbers below the detection limit as it requires those samples to be enriched before detection. Estimation of the diagnostic sensitivity and specificity for A. laidlawii was not possible given that none of the study samples tested positive for A. laidlawii by laboratory culture. In addition, estimates of diagnostic accuracy for Mycoplasma spp. and Mycoplasma bovis reported here should be interpreted with caution given the small sample size used in their diagnostic test evaluation.

Conclusions

Because the multiplex assay results in amplifying all three target DNA sequences within the same tube, the costs, labor, and time required for sample preparation are reduced. This assay therefore provides an opportunity for more rapid diagnosis of contagious Mycoplasma pathogens and the reduction of risk for false positive results caused by A. laidlawii. In particular, the assay can be used as a first pass identification method for making preliminary decisions on mycoplasma diagnosis, thereby minimizing the risk of pathogen spread within dairy herds.

Supplemental Information

Supplemental Information 1 Multiple sequence alignments of 16S rRNA (118 bp), rpoB (232 bp) and I6S-23S ITS (108 bp) DNA sequences targeting the Mycoplasma genus, M. bovis and A. laidlawii, respectively.

Positions identical to the first sequence are indicated by dots and gaps indicated by dashes. The primers and probe (reverse compliment) binding sequences are underlined and highlighted. The rpoB, 16S rRNA genes, and ITS regions used for alignment are from the following strains: M. bovis strain PG45 (ATCC 25523, NC_014760), M. californium strain ST-6 (ATCC 33461, NZ_CP007521), M. bovigenitalium strain HAZ (ATCC 19852, AP017902), M. canadense strain HAZ 360_1 (ATCC 29410, NZ_AP014631), M. alkalescens 14918 (ATCC 29103, NZ_AMWK01000000) and A. laidlawii PG-8A (ATCC 23206, NC_010163)

Click here for additional data file.

Supplemental Information 2 Standard curves for the 16S rRNA, rpoB and 16S-23S ITS TaqMan assays preformed individually (singleplex).

The standard curves were constructed with 10-fold serial dilutions of M. bovis ATCC 25523 (16S rRNA and rpoB assays), M. bovigenitalium ATCC 19852 (16S rRNA assay), M. californicum ATCC 33461 (16S rRNA assay), and A. laidlawii ATCC 23206 (ITS assay) gDNA, ranging from between approximately 5 fg to 5 ng gDNA. Results shown are from a single run with each dilution tested in triplicate. The R2 value was 0.99 for the standard curve of each target (16S rRNA, rpoB and ITS). Error bars indicate standard deviation (±) based on the results for three replicates.

Click here for additional data file.

Supplemental Information 3 qPCR standard curves.

Raw data files of singplex and multiplex assays, limit of detection determination in milk, and mixed-species qPCR detection

Click here for additional data file.

Additional Information and Declarations

Competing Interests

Author Contributions

Data Availability

Sharif S. Aly is an Academic Editor for PeerJ.

Kanika Chauhan performed the experiments, analyzed the data, prepared figures and/or tables, authored or reviewed drafts of the paper, and approved the final draft.

Sharif S. Aly conceived and designed the experiments, analyzed the data, authored or reviewed drafts of the paper, and approved the final draft.

Terry W. Lehenbauer conceived and designed the experiments, authored or reviewed drafts of the paper, and approved the final draft.

Karen H. Tonooka performed the experiments, analyzed the data, authored or reviewed drafts of the paper, and approved the final draft.

Kathy Glenn performed the experiments, analyzed the data, authored or reviewed drafts of the paper, and approved the final draft.

Paul Rossitto performed the experiments, authored or reviewed drafts of the paper, and approved the final draft.

Maria L. Marco conceived and designed the experiments, analyzed the data, prepared figures and/or tables, authored or reviewed drafts of the paper, and approved the final draft.

The following information was supplied regarding data availability:

The data are available in the Supplemental Files.

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
