# Peer review of "Development of a multiplex qPCR assay for the simultaneous detection of Mycoplasma bovis, Mycoplasma species, and Acholeplasma laidlawii in milk"

_PeerJ, doi:10.7717/peerj.11881_

## Round 0.1 · original submission · Major Revisions

Dear Dr. Chauhan and colleagues:

Thanks for submitting your manuscript to PeerJ. I have now received three independent reviews of your work, and as you will see, the reviewers raised some concerns about the research. Despite this, these reviewers are optimistic about your work and the potential impact it will have on research studying diagnostic approaches for detecting pathogens in milk. Thus, I encourage you to revise your manuscript, accordingly, taking into account all of the concerns raised by both reviewers.

While the concerns of the reviewers are relatively minor, this is a major revision to ensure that the original reviewers have a chance to evaluate your responses to their concerns. There are many suggestions, which I am sure will greatly improve your manuscript once addressed.

I look forward to seeing your revision, and thanks again for submitting your work to PeerJ.

Good luck with your revision,

-joe

Reviewer 1 ·

Basic reporting

Very well written and presented manuscript.

Experimental design

Clear aims
Methods are appropriate, detailed and thorough.

Validity of the findings

Conclusions are well supported by the data.

Additional comments

Reviewing Manuscript 58967v1
Development of a multiplex qPCR assay for the simultaneous detection of Mycoplasma bovis, Mycoplasma species, and Acholeplasma laidlawii in milk

This manuscript describes the development and application of a multiplex PCR assay for detection of 3 target bacterial groups in milk. These include the pathogens Mycoplasma spp and Mycoplasma bovis and environmental contaminant Acholeplasma laidlawaii. qPCR optimization, efficiency and detection limits were performed and specificity with varying levels of bacterial targets was confirmed. The method was used to test a number of milk samples and shown to have very good sensitivity and specificity. The assay offers several advantages over previously developed qPCR methods, including sensitivity and usefulness of the targets detected. The manuscript is very well written and the findings are of high quality.

Specific comments:
- Line 178 – primer and probe concentration ranges look different here compared to that in Table 2 (75-200nM)
- Line 220 – Table number (4?) missing
- Line 335 – (Ct ‘greater than or equal to’ symbol should be ‘less than or equal to’)
- Table 2 – it would be good to include the full names of ‘VIC’, ‘QSY’, ‘ABY’ and ‘FAM’ in the footnote of this table

Reviewer 2 ·

Basic reporting

There are few editorial changes required better attention to details. Some examples where the text could be improved include:
Line 43-44: suggest replacing Mycoplasma bovis with M. bovis, the abbreviation was introduced on Line 30.
Line 85: suggest replacing Mycoplasma sp. with Mycoplasma spp.
Line 141, 144, 150: suggest “The gDNA”.
Line 220: A number of the table is missing.

Experimental design

This paper describes laboratory and field performance validation of a quantitative multiplex PCR assay for the detection of Mycoplasma species, Mycoplasma bovis and Acholeplasma laidlawii. Fast detection and identification of these pathogens can greatly benefit prevention and control of bovine mastitis caused by M. bovis and Mycoplasma spp. The field milk samples used for the validation was not possible to validate the test for A. laidlawii, aside for confirmation of absence.

Overall the study is well described and the manuscript generally well written. There are minor points that require clarification (listed below) and I would be grateful if the authors could consider these.

Line 212-220: The authors prepared two separate gDNA samples from raw milk of which M. bovis and A. laidlawii were separately inoculated. The two gDNAs were then mixed in equal volumes (1:1 ratios), is there a particular reason for this? To mimic a field milk sample, a mixer of milk with equal amount of both inoculated strains could be used for DNA extraction, was this considered?

Line 222-236: The paper describes testing of field milk samples by a multiplex qPCR assay compared to a reference method (cultivation). The authors could consider adding a Cohen's kappa coefficient to determine agreement between two assays.

Line 227: What is an evidence of growth? Perhaps provide a suitable detail or criteria.

Validity of the findings

Line 245: The author described “testing between 75 and 200 nM of each primer and TaqMan probe, different Taq polymerase reagents, and several annealing temperatures (data not shown)”. If the assays are important and included in the manuscript, the authors could consider adding the results. There is currently no mention of results on the different parameter tests in the manuscript. The authors should consider adding results or revise the paragraph on line 176-187.

Line 290-291: The authors reported that milk alone resulted in DNA amplification with Ct 33 to 37. Could the authors please add a comment in this regard to discussion?

Reviewer 3 ·

Basic reporting

Dear Mr. Editor,

First I apologize for my delay due to several prior engagements as well as COVID 19 pandemic.

My general observations are:

Article is quite lengthy. Authors wish to say about the development of multiplex qPCR for the detection of M. bovis, Mycoplasma spp. (which they should name in Abstract as well as in Introduction) and A. laidlawii. I could not understand that when they are targeting pathogenic etiology of mollicutes for Bovine contagious mastitis (although this is not a standard terminology for Bovine mastitis), then they should exclude A. laidlawii.

Molecular detection methods always be sharply targeted for specific antigen.

The article is quite lengthy so should be re written to make it short. I feel that in spite of expressing their research work, they have rather justified their work with elaborated text.
I fear that this may cause "loosing of interest" of our valuable readers.

In title also they wrote SIMULTANEOUS DETECTION. It is well known that multiplexes are always SIMULTANIOUS, so they should remove this word from title.

Experimental design

Experimental design is Good.

Authors should tell about the superiority of this qPCR in terms of their targeted gene, in terms of COPY NUMBER of the detectable gene / CFUs.

If primers are novel then they must emphasise this somewhere in text.

Validity of the findings

They have duly validated their findings with several other bacterial / mollicutes isolates as well as standard cultures.

Additional comments

1. Reduce the words of manuscript.
2. Remove word "simultaneous" from the title.
3. Line no. 24: name all the mycoplasma species.
4. Line 43: Remove the word "help". Be specific and sharp in your text.
5. Authors should write about the Comparison of their developed Multiplex with other PRE PUBLISHED qPCR papers, beside giving emphasis to their Diagnostic Sensitivity and Diagnostic specificity.
6. Line no. 113: replace word "from" with "of".
7. Line 139: Under title "Genomic DNA extraction", give reference of gDNA extraction from milk as milk contains fatty globules and other salts, to it is always difficult to get good quantity of gDNA from milk. Readers would like to know about the reference.
8. Line 183: Remove " : " after "with".
9. Line no. 432 - 433: "In future, methods that accelerate DNA extraction .......Mycoplasma in milk"; what authors want to suggest. So rewrite the sentence or remove this. I feel that such suggestions are of INDEFINATE type and should be avoided.

---

## Round 0.2 · accepted · Accept

Dear Dr. Chauhan and colleagues:

Thanks for revising your manuscript based on the concerns raised by the reviewer. I now believe that your manuscript is suitable for publication. Congratulations! I look forward to seeing this work in print, and I anticipate it being an important resource for groups studying diagnostic approaches for detecting pathogens in milk. Thanks again for choosing PeerJ to publish such important work.

Best,

-joe

Reviewer 1 ·

Basic reporting

NA

Experimental design

NA

Validity of the findings

NA

Additional comments

My suggested changes have been implemented. The only thing I noticed was in Table 2, the concentration of primers/probe for A. laidlawii are out of alignment.